# An Exploration of Resilience and Positive Affect among Undergraduate Nursing Students: A Longitudinal Observational Study

L. Iván Mayor-Silva [1] , Alfonso Meneses-Monroy [1] , Leyre Rodriguez-Leal [2,*] and Guillermo Moreno [1,3,*]

1   Departamento de Enfermería, Facultad de Enfermería, Fisioterapia y Podología, Universidad Complutense de Madrid, 28040 Madrid, Spain; limayors@ucm.es (L.I.M.-S.); ameneses@ucm.es (A.M.-M.)
2   Red Cross Nursing University College, Autonomous University of Madrid, 28003 Madrid, Spain
3   Grupo de Investigación Cardiovascular Multidisciplinar Traslacional (GICMT), Área de Investigación Cardiovascular, Instituto de Investigación Hospital 12 de Octubre (imas12), 28041 Madrid, Spain
*   Correspondence: leyre.rodriguez@cruzroja.es (L.R.-L.); guimoren@ucm.es (G.M.)

**Abstract:** Background: The purpose of this study is to analyze the variation in resilience and emotional state scores in nursing students throughout the four years of training for the nursing degree. Methods: This is a longitudinal observational study of a paired and prospective cohort of 176 nursing students who enrolled in the first year of a bachelor's degree in 2019. The study followed up with the students in 2022 and examined several sociodemographic factors, including sex, marital status, date of birth, living arrangements and occupation. Additionally, the study investigated changes in negative affect, positive affect, and resilience. Results: A total of 176 students participated in the study. The study found that resilience increased from $68.24 \pm 10.59$ to $70.87 \pm 9.06$ ($p < 0.001$), positive affect increased from $28.16 \pm 4.59$ to $33.08 \pm 8.00$ ($p < 0.001$), and the negative affect score decreased from $25.27 \pm 5.12$ to $21.81 \pm 7.85$ ($p < 0.001$). The study also found that married individuals experienced an increase in negative affect ($p = 0.03$) compared to singles or those in open relationships. Furthermore, the change in resilience was greater in men than in women ($p = 0.01$). Conclusions: Throughout their four-year training, nursing students experience an increase in resilience and positive affect, as well as a decrease in negative affect.

**Keywords:** resilience; positive affect; negative affect; nursing students

## 1. Introduction

Resilience and affectivity are two psychological constructs that are highly relevant in the educational field of nursing students, playing a crucial role in their well-being and academic performance [1,2]. Resilience is defined as the ability to cope with adverse situations and is a dynamic process that involves the interaction of individual, social, and environmental factors [3]. Foundational skills for maintaining a healthy state of mind are crucial for meeting the challenges inherent in the nursing profession [4].

Resilience is considered a crucial competency in nursing, as it is related to three diagnoses in the NANDA taxonomy [5]. Furthermore, research suggests that resilience is a necessary characteristic for nurses due to their continuous exposure to human suffering and stressful working conditions [6,7]. Nurses can use resilience to support and motivate patients in coping with the challenges of illness while enhancing their own skills and resources to remain resilient. In addition, higher levels of resilience help them maintain their own health and well-being, ensuring that the physical, emotional, and mental demands of caregiving do not overwhelm them or exhaust their ability to provide care [8]. Resilience can also reduce the symptoms of burnout and its consequences in the nursing profession [9]. Resilience plays a key role in strengthening nursing students by allowing them to bounce back from challenging situations. The range of experiences, whether encouraging or

discouraging, profoundly influences their outlook and confidence throughout their clinical education [10].

Affectivity pertains to an individual's subjective emotional state at a particular moment. It is worth noting that affectivity has a significant correlation with an individual's perceived well-being and discomfort [11]. Affective states can range from positive moods, such as happiness and contentment, which increase motivation, satisfaction, and interest in activities, to negative states, such as sadness and anxiety, which generate disinterest, lack of energy, and lack of pleasure in life. The affectivity of nursing students can be influenced by various factors, including academic workload, clinical demands, and emotional experiences associated with patient care [12].

Various studies have explored the relationship between resilience and affectivity. Specifically, in nursing students, resilience can influence how they cope with the academic and emotional challenges associated with their professional training. Several studies have shown a positive correlation between resilience scores and positive affect scores [13–15]. This indicates that nursing students with higher levels of resilience tend to experience more positive emotions. Previous research has explored the correlation between resilience, mood, and stress. The findings suggest that nursing students with higher levels of resilience experience fewer negative emotions, indicating a greater ability to cope with stress and adversity. These results support the notion that resilience is linked to higher positive affect and lower negative impact, which could have significant implications for the well-being and performance of nursing students. For this reason, some authors recommend resilience training starting at the undergraduate level due to its importance for future professionals [16–18].

In the past five years, Spanish universities have implemented various activities to enhance student resilience. These include telephone psychological support programs, clinical simulations, peer mentoring programs, student advocacy, clinical practices with tutors, and participatory conferences on the profession to improve communication among classmates. These strategies have also been incorporated into the formal curriculum at our local university over the past 10 years through participation in clinical programs, simulations, and peer mentoring. The aim of this study is to analyze the variation in resilience and emotional state scores in nursing students throughout the 4 years of training within the nursing degree.

## 2. Material and Methods

### 2.1. Design

From September 2019 to October 2022, we conducted a single-center, longitudinal observational study of paired and prospective cohorts using the STROBE statement as a framework [19].

### 2.2. Participants

The study population comprised 224 nursing students enrolled in the first year of the bachelor's degree in 2019. The researchers used convenience sampling to recruit students through online messages in the academic electronic platform used by professors to communicate with students (virtual campus) in September 2019 and followed up with them in October 2022. None of the students' professors were involved in this study. The data analysis included all students who had completed both assessments.

### 2.3. Study Variables

Study variables included sociodemographic factors such as sex (male/female), marital status (married/single/open relationship), age, living arrangement (with friends or fellow students/with parents or relatives/as a couple/alone/none of the above), and occupation (sporadic professional activities/continuous and remunerated work/volunteering/none of the above), as well as negative affect, positive affect, and resilience, which were assessed through self-administered online questionnaires sent through the virtual campus:

- The Positive and Negative Affect Schedule (PANAS) [20] questionnaire is a self-report instrument used to evaluate an individual's affective state [11]. It consists of two sub-scales, each comprising 10 items that are rated on a Likert-type scale with five response options ranging from 1 (little or almost nothing) to 5 (extremely). One scale measures recent positive experiences, which act as a protective element, while the other measures negative experiences, which act as a risk factor for diseases [21]. The scales have demonstrated consistency and stability in the Spanish university population, with good construct validity and a reliability index of Cronbach's alpha >0.87 [22].
- The study collected the Connor–Davidson Resilience Scale (CD-RISC) [23] in the Spanish version [24], which measures resilience using 25 items in a Likert-type response format with five response options (0 = 'not at all', 1 = 'rarely', 2 = 'sometimes', 3 = 'often', and 4 = 'almost always'). The scale ranges from 0 to 100, with higher scores indicating greater resilience. The items are grouped into five dimensions: Persistence, Tenacity, and Self-Efficacy; Control under Pressure; Adaptability and Networking; Control and Purpose; and Spirituality. The thresholds are less than 70 (low), 70 to 87 (intermediate), and greater than 88 (high). The Spanish version has optimal internal consistency, with a Cronbach's alpha of 0.86 [24].

### 2.4. Data Analysis

Numerical codes were used to match and anonymize participants' personal information to ensure confidentiality. Relative and absolute frequencies were calculated for qualitative variables, while measures of central tendency and dispersion were calculated for continuous variables, with 95% confidence intervals. The normality of the variables was assessed using the Kolmogorov–Smirnov test. Bivariate relationships between sociodemographic variables and the main variables were analyzed using ANOVA and Student t tests, both paired and for independent samples. A statistically significant relationship was recognized when the *p*-value was less than 0.05. Finally, a correlation analysis was conducted between the main variables of the study. Data analysis was performed using SPSS version 26®.

### 2.5. Ethical Considerations

This study adhered to the ethical principles for medical research and maintained the confidentiality of data in accordance with Regulation (EU) 2016/679 of the European Parliament and of the Council of 27 April 2016 on Data Protection (GDPR). The Declaration of Helsinki on Biomedical Research Involving Human Subjects was also followed throughout the study. The research team obtained written consent from each student outside of class time, clearly explaining the voluntary and anonymous nature of the study. The project was approved by the Research Committee of the center (approval number: FEFP 20/21).

## 3. Results

A total of 176 first-year nursing students participated in the study, resulting in a response rate of 78.5%. After four years, 97.15% of the students (*n* = 171) returned to participate, resulting in a loss of follow-up of only 2.85%. Table 1 displays the sociodemographic characteristics of the sample at the beginning of the study, with 30 (17.5%) male and 145 (82.5%) female participants. The age range was between 21 and 57 years, with a mean age of 23.14 years (SD: 3.97). Out of the participants, 147 (84.8%) were single, 134 (78.4%) did not engage in any work or extracurricular activities outside of the university, and 147 (86.0%) lived with their parents or relatives.

**Table 1.** Sociodemographic characteristics of the sample at the beginning of the study (2019–2020 academic year, *n* = 176).

| | | n | % | Mean | SD |
|---|---|---|---|---|---|
| **Sex** | Male | 30 | 17.5 | | |
| | Female | 141 | 82.5 | | |
| **Age** | | | | 23.14 | 3.98 |
| **Marital status** | Married | 5 | 2.9 | | |
| | Single | 145 | 84.8 | | |
| | Open relationship | 21 | 12.3 | | |
| **Living arrangements** | With friends or fellow students | 13 | 7.6 | | |
| | With parents or relatives | 147 | 86.0 | | |
| | As a couple | 6 | 3.5 | | |
| | Alone | 4 | 2.3 | | |
| | None of the above | 1 | 0.6 | | |
| **Occupation** | Sporadic professional activities | 35 | 20.5 | | |
| | Volunteering | 15 | 8.8 | | |
| | Continuous and remunerated work | 33 | 19.3 | | |
| | None of the above | 88 | 51.54 | | |

*3.1. Variation in Resilience*

The CD-RISC questionnaire produced an average score of 68.24 (SD: 10.59) points in the first year, which increased to an average of 70.87 (SD: 9.06) points in the fourth year ($p < 0.001$). In the first year, only 0.6% ($n = 1$) of the students exhibited high resilience, while 44.4% ($n = 76$) exhibited intermediate resilience, and 55% ($n = 94$) exhibited low resilience. In the fourth year, 2.9% ($n = 5$) of the students demonstrated a high level of resilience, while 48.5% ($n = 83$) and 48.5% ($n = 83$) demonstrated intermediate and low levels of resilience, respectively. Of the students, 20.5% ($n = 35$) showed an increase in resilience level (from low to intermediate or high resilience and from intermediate to high resilience), 66.7% ($n = 114$) maintained their resilience level, and 12.9% ($n = 22$) showed a decrease in resilience level (from high or intermediate to intermediate or low). This change was statistically significant ($p < 0.001$).

The average score for personal competence increased significantly from 16.44 (SD: 3.77) points in the first year to 17.48 (SD: 2.95) points in the fourth year ($p < 0.001$). Similarly, the average score for tolerance of negative affect and the strengthening effects of stress increased from 16.60 (SD: 3.88) points in the first year to 17.75 (SD: 3.38) points in the fourth year ($p < 0.001$). Positive acceptance of change and secure relationships obtained an average score of 15. The study found that in the first year, the score was 15.70 (SD: 2.78) points, which decreased to 15.66 (SD: 2.27) points in the fourth year ($p = 0.87$). The average score for a sense of control was 8.63 (SD: 2.16) points in the first year, which increased to 9.16 (SD: 1.58) points in the fourth year ($p < 0.001$). Spiritual influence had an average score of 4.30 (SD: 2.20) points in the first year, which decreased to 4.22 (SD: 2.06) points in the fourth year ($p < 0.001$). Table 2 displays the change in specific items.

**Table 2.** Results of the CD-RISC questionnaire by items and dimensions in the first and fourth year.

| | 1st Year | | 4th Year | | Change | | | |
|---|---|---|---|---|---|---|---|---|
| | **Mean** | **SD** | **Mean** | **SD** | **Mean** | **SD** | **t *** | *p* ** |
| **CD-RISC items** | | | | | | | | |
| 1. I'm able to adapt when changes occur | 3.00 | 0.89 | 2.94 | 0.70 | −0.06 | 1.01 | 0.84 | 0.40 |
| 2. I have one close and secure relationship | 3.39 | 0.81 | 3.39 | 0.64 | −0.01 | 0.86 | 0.09 | 0.93 |
| 3. Sometimes fate or God helps me | 1.43 | 1.35 | 1.57 | 1.23 | 0.15 | 1.16 | −1.65 | 0.10 |
| 4. I can deal with whatever comes my way | 2.86 | 0.81 | 2.84 | 0.68 | −0.02 | 0.85 | 0.36 | 0.72 |
| 5. Past successes give me confidence | 3.34 | 0.87 | 3.36 | 0.80 | 0.02 | 1.01 | −0.30 | 0.76 |
| 6. I try to see the humorous side of things | 2.68 | 1.00 | 3.00 | 0.80 | **0.32** | 1.01 | −4.17 | **<0.001** |
| 7. Having to cope with stress makes me stronger | 2.71 | 1.05 | 2.74 | 0.88 | 0.03 | 1.22 | −0.31 | 0.75 |
| 8. I tend to bounce back after illness, injury or other hardships | 3.11 | 0.87 | 3.14 | 0.73 | 0.04 | 1.01 | −0.45 | 0.65 |
| 9. I believe most things happen for a reason | 2.88 | 1.32 | 2.64 | 1.21 | **−0.23** | 1.22 | 2.50 | **0.01** |
| 10. I make my best effort, no matter what | 3.25 | 0.80 | 3.18 | 0.69 | −0.08 | 0.88 | 1.13 | 0.26 |
| 11. I believe I can achieve my goals, even if there are obstacles | 3.08 | 0.79 | 3.11 | 0.59 | 0.02 | 0.88 | −0.35 | 0.73 |
| 12. Even when hopeless, I do not give up | 2.89 | 0.90 | 2.95 | 0.73 | 0.06 | 1.05 | −0.80 | 0.43 |
| 13. In times of stress, I know where to find help | 2.97 | 1.09 | 3.19 | 0.81 | **0.22** | 1.16 | −2.51 | **0.01** |
| 14. Under pressure, I stay focused and think clearly | 2.26 | 1.19 | 2.63 | 0.94 | **0.36** | 1.24 | −3.82 | **<0.001** |
| 15. I prefer to take the lead in problem-solving | 2.71 | 1.12 | 2.91 | 0.81 | **0.20** | 1.06 | −2.52 | **0.01** |
| 16. I am not easily discouraged by failure | 1.74 | 1.17 | 2.24 | 0.90 | **0.50** | 1.31 | −5.03 | **<0.001** |
| 17. I think of myself as a strong person when dealing with life's changes and difficulties | 2.40 | 1.17 | 2.83 | 0.89 | **0.43** | 1.10 | −5.14 | **<0.001** |
| 18. I make unpopular or difficult decisions | 2.19 | 0.91 | 2.18 | 0.86 | −0.01 | 0.98 | 0.16 | 0.88 |
| 19. I am able to handle unpleasant or painful feelings like sadness, fear and anger | 2.49 | 1.01 | 2.65 | 0.79 | **0.16** | 1.02 | −2.10 | **0.04** |
| 20 I have to act on a hunch | 1.57 | 1.01 | 1.65 | 0.88 | 0.09 | 1.13 | −1.01 | 0.31 |
| 21. I have a strong sense of purpose in life | 2.88 | 0.85 | 3.09 | 0.62 | **0.20** | 1.00 | −2.68 | **0.01** |
| 22. I feel like I am in control | 2.78 | 0.99 | 2.88 | 0.75 | 0.10 | 1.09 | −1.19 | 0.23 |
| 23. I like challenges | 2.73 | 1.01 | 2.78 | 0.86 | 0.05 | 0.93 | −0.74 | 0.46 |
| 24. I work to attain goals | 3.54 | 0.67 | 3.49 | 0.57 | −0.05 | 0.78 | 0.89 | 0.38 |
| 25. I take pride in my achievements. | 3.44 | 0.85 | 3.50 | 0.68 | 0.06 | 0.80 | −0.95 | 0.34 |
| **CD-RISC dimensions** | | | | | | | | |
| Personal competence | 16.44 | 3.77 | 17.48 | 2.95 | **2.61** | 3.23 | −4.03 | **<0.001** |
| Tolerance of negative affect and the strengthening effects of stress | 16.60 | 3.88 | 17.75 | 3.38 | **3.00** | 3.73 | −4.23 | **<0.001** |
| Positive acceptance of change and secure relationships | 15.70 | 2.78 | 15.66 | 2.27 | 2.05 | 2.47 | 0.16 | 0.87 |
| Sense of control | 8.63 | 2.16 | 9.16 | 1.58 | **1.40** | 1.78 | −2.87 | **<0.001** |
| Spiritual influence | 4.30 | 2.20 | 4.22 | 2.06 | 1.87 | 2.23 | 0.68 | 0.50 |

* Note: Student t test for paired samples. ** Note: Significant results (*p* < 0.05).

### 3.2. Variation of Affective State at the Beginning and End of the Degree

The PANAS questionnaire revealed a statistically significant increase in positive affect score from a mean of 28.16 (SD: 4.59) points in year one to a mean of 33.08 (SD: 8.00) points in year four (*p* < 0.001). Additionally, the negative affect score decreased from a mean of 25.27 (SD: 5.12) in the first year to 21.81 (SD: 7.85) in the fourth year (*p* < 0.001). Table 3 presents a comparison of specific items from the PANAS questionnaire in the first and fourth years. The data indicates a statistically significant increase in interest, distress, excitement, upset, strength, pride, alertness, and attention and a decrease in guiltiness, hostility, nervousness, and jitteriness.

**Table 3.** Results of the PANAS questionnaire in first and fourth years and their change.

| | 1st Year | | 4th Year | | Change | | | |
| --- | --- | --- | --- | --- | --- | --- | --- | --- |
| | **Mean** | **SD** | **Mean** | **SD** | **Mean** | **SD** | **t \*** | **p \*\*** |
| **Interested** | 3.27 | 0.92 | 3.69 | 0.97 | **0.41** | 1.24 | −4.42 | **<0.001** |
| **Distressed** | 2.49 | 1.2 | 2.9 | 1.21 | **0.86** | 1.514 | 7.51 | **<0.001** |
| **Excited** | 2.93 | 1.07 | 3.45 | 0.99 | **0.44** | 1.28 | 4.49 | **<0.001** |
| **Upset** | 2.23 | 1.06 | 2.47 | 1.21 | **0.24** | 1.53 | 2.08 | **0.04** |
| **Strong** | 3.16 | 0.97 | 3.44 | 1.06 | **0.26** | 1.23 | 1.96 | **0.01** |
| **Guilty** | 1.88 | 1.17 | 1.65 | 0.98 | −0.23 | 1.55 | 6.30 | 0.05 |
| **Scared** | 1.95 | 1.03 | 2.03 | 1.11 | 0.10 | 1.41 | 0.97 | 0.33 |
| **Hostile** | 1.97 | 0.99 | 1.65 | 0.97 | **−0.32** | 1.42 | −2.94 | **<0.001** |
| **Enthusiastic** | 3.44 | 0.93 | 3.36 | 1.09 | −0.09 | 1.29 | −0.88 | 0.38 |
| **Proud** | 3.37 | 0.94 | 3.78 | 1.11 | **0.41** | 1.37 | 3.92 | **<0.001** |
| **Irritable** | 2.39 | 1.07 | 2.51 | 1.15 | 0.12 | 1.36 | 1.17 | 0.24 |
| **Alert** | 3.02 | 1.01 | 3.29 | 0.97 | **0.28** | 1.27 | 2.86 | **0.01** |
| **Ashamed** | 1.78 | 1.02 | 1.62 | 0.92 | 0.17 | 1.21 | −1.82 | 0.07 |
| **Inspired** | 2.74 | 0.98 | 2.9 | 1.03 | 0.15 | 1.33 | 1.48 | 0.14 |
| **Nervous** | 3.25 | 1.11 | 2.94 | 1.22 | **−0.30** | 1.53 | −2.58 | **0.01** |
| **Determined** | 3.21 | 0.97 | 3.3 | 0.93 | 0.08 | 1.10 | 0.96 | 0.34 |
| **Attentive** | 3.01 | 0.94 | 3.18 | 0.92 | **−0.20** | 1.21 | −2.12 | **0.04** |
| **Jittery** | 1.92 | 1.07 | 2.28 | 1.1 | **−0.20** | 1.52 | 0.09 | **<0.001** |
| **Active** | 3.39 | 0.93 | 3.52 | 0.96 | 0.13 | 1.25 | 1.40 | 0.16 |
| **Afraid** | 2.03 | 1.08 | 2.02 | 1.12 | 0.06 | 1.28 | 0.65 | 0.52 |

\* Note: Student t test for paired samples. \*\* Note: Significant results ($p < 0.05$).

### 3.3. Changes According to Sociodemographic Characteristics

Table 4 shows statistically significant differences in the change of negative affect. It is observed that married individuals experience an increase in negative affect ($p = 0.03$). Furthermore, the change in resilience is greater in men than in women ($p = 0.01$). No statistically significant differences were observed in the remaining sociodemographic variables.

**Table 4.** Change in resilience and positive and negative affect according to sociodemographic variables.

| | | Change Positive Affect | | | Change Negative Affect | | | Change CD-RISC | | |
| --- | --- | --- | --- | --- | --- | --- | --- | --- | --- | --- |
| | | **Mean** | **SD** | **p \*** | **Mean** | **SD** | **p \*** | **Mean** | **SD** | **p \*** |
| **Sex** | Man | 6.83 | 8.90 | 0.15 | −4.00 | 8.63 | 0.85 | 6.57 | 11.12 | **0.01 \*\*** |
| | Woman | 4.36 | 8.42 | | −3.68 | 8.27 | | 1.72 | 9.37 | |
| **Marital status** | Married | 3.40 | 13.16 | 0.91 | 3.80 | 8.41 | **0.03 \*\*** | −4.20 | 11.45 | 0.26 |
| | Single | 4.78 | 8.57 | | −4.36 | 8.17 | | 2.92 | 9.63 | |
| | Open relationship | 5.24 | 7.42 | | −1.24 | 8.28 | | 1.76 | 10.79 | |
| **Living arrangements** | With friends or fellow students | 5.69 | 9.93 | 0.53 | −3.15 | 8.37 | 0.71 | 6.92 | 7.29 | 0.17 |
| | With parents or relatives | 4.80 | 8.44 | | −3.80 | 8.09 | | 2.52 | 9.60 | |
| | As a couple | 0.50 | 10.03 | | −1.67 | 15.11 | | 1.50 | 17.03 | |
| | Alone | 9.50 | 4.80 | | −8.00 | 4.55 | | −6.50 | 10.08 | |
| | None of the above | 0.00 | 0.00 | | 3.00 | 0.00 | | −4.00 | 0.00 | |
| **Occupation** | Sporadic professional activities | 4.00 | 7.84 | 0.89 | −2.66 | 8.16 | 0.66 | 3.63 | 11.32 | 0.71 |
| | Continuous and remunerated work | 4.76 | 10.87 | | −5.18 | 9.48 | | 2.21 | 9.29 | |
| | Volunteering | 4.13 | 6.40 | | −3.67 | 6.69 | | 0.13 | 9.95 | |
| | None of the above | 5.24 | 8.23 | | −3.64 | 8.21 | | 2.69 | 9.48 | |

\* Note: One-factor ANOVA. \*\* Note: Significant results ($p < 0.05$)

## 4. Discussion

The aim of this study was to investigate changes in resilience and affect during the four-year bachelor's degree in nursing program to determine if the students improved in these areas for their future clinical practice. It is important to note that this study focused solely on resilience and affect and did not investigate other factors that may impact student success. Our findings indicate that 55% of students begin the program with low resilience, which is consistent with the findings of other authors [2]. Over the course of the four-year program, just over 20% of students demonstrated an increase in their resilience score. The data generally indicate that students exhibit greater resilience, although the most significant variations were observed between low and intermediate resilience levels without reaching high scores. Therefore, there is still room for improvement. In addition, compared to their resilience levels in the first year, some students showed a decrease in resilience. This fact is significant because various studies have demonstrated that the university population generally exhibits medium to high levels of resilience [16,25].

In terms of resilience, personal competence, tolerance of negative affect, and the strengthening effects of stress and sense of control, there is an increase during the four-year period. This may be a normal aspect of academic development, given the complexity of the degree, the difficulty of the subjects, the practical clinical training, and the need to balance work and personal life. The results indicate that the academic and clinical training process provided during the bachelor's degree in nursing can contribute to the development of resilience in students, similar to other professions [26].

However, this decrease in positive acceptance of change and secure relationships, as well as spiritual influence, is negative since these factors are the ones that most protect nursing professionals against burnout syndrome and other difficult situations typical of the nursing role [27,28], such as post-traumatic stress [29]. Several studies have associated spiritual influence with resilient behaviors, which can improve quality of life and act as a protective factor against suicidal ideation [30,31]. Spiritual influence also appears to be a good modulator of the effects of burnout on mental health [32]. Additionally, extroversion and secure relationships have been found to be associated with greater satisfaction with nursing work [33].

Although hospital practice can contribute to the development of nursing students' resilience, studies have shown that their resilience scores are moderate overall [34]. This work is consistent with those findings. This shows that these difficult situations posed by clinical practice will strengthen the student as long as there is support from the tutor, support from peers, and better management of these experiences by the student [15,35–37].

It is important to consider that the students may have been impacted by the COVID-19 pandemic in both positive and negative ways, affecting their resilience. Additionally, the social environment before the pandemic may differ greatly from the post-pandemic world, particularly in areas such as technology, social relationships, and personal fulfillment [38,39]. Therefore, the variations observed in the scores may be attributed to this event and the resulting social change. In the post-pandemic era, it is important to implement pedagogical interventions that have been proven effective in developing resilience in nursing students [40,41] to prevent future risks to the mental health of future professionals.

In terms of affect, positive affect increased while negative affect decreased over the four-year period. The observed changes included an increase in distress, excitement, upset, strength, pride, alertness, and attention, as well as a decrease in guiltiness, hostility, nervousness, and jitteriness. These changes may act as protective or risk factors in the development of diseases among new professionals. These findings may reflect the fact that, overall, nursing students are generally satisfied with their clinical learning experiences [42] and demonstrate that students are positively influenced by professors and their clinical practices, as other studies have suggested [43]. Despite evidence suggesting that confinement during COVID-19 had a negative impact on mood [39], our students were able to increase positive affect and decrease negative affect by the end of their training period, indicating that the influence of confinement was mitigated in the following years.

The study has shown that sex and marital status are factors that affect the development of these variables. Single individuals experience a greater increase in negative affect compared to married individuals or those in an open relationship. Some studies suggest that marriage may have a positive impact on well-being by reducing negative effects in the short term. However, this beneficial effect may diminish in the long term, particularly for women [44]. This could explain why negative affect changes more in married couples compared to single or open relationships. Our findings indicate that men experience a greater increase in resilience levels than women. Other studies on students have found results similar to ours, where men exhibit significantly higher levels of resilience than women. In contrast, women score higher in their perception of stress and social support [45]. Male adolescents scored significantly higher in resilience than females, which may be related to the brain maturing as adolescents grow [46]. Men and women cope differently with adversity. This may be because certain resilience factors are more effective for one gender than the other [47]. Exposure to stress during puberty tends to have more immediate and significant consequences for girls, heightening the likelihood of mood and stress disorders like depression, anxiety, and PTSD. The hormonal shifts that occur with menopause influence emotional regulation in women. Moreover, early life hardships can also modify the influence of estradiol on brain functions [48].

Although no other sociodemographic variable has been found to contribute to influence resilience and affect, other studies have reported that self-esteem, family support, subjective well-being, psychological well-being [49], regular sleep, perceived stress, well-being [50], empathy, responsibility, optimism, hope and interpersonal skills (communication, self-efficacy, self-control, autonomy, problem-solving) [51] may influence resilience levels. On the other hand, perceived teaching quality and professors' communication technology competence [52], self-regulatory capacity (boredom, awareness, goal, and emotion control) [53], self-efficacy [54], bedtime procrastination, and mobile phone addiction [55] may influence positive and negative emotions in students. Future studies should include these and new factors to control for changes in resilience and positive/negative effects over time.

### 4.1. Practical Implications

Based on our main findings, it is important to note that the demands of nursing education, such as schedules, classes, and clinical practice, may not be adequate for developing the required level of resilience for future professional practice. In fact, this approach may even diminish resilience factors. Our findings suggest that promoting support from colleagues and tutors, teaching coping skills, and emphasizing the significance of spirituality could lead to more resilient professionals. This is important to attain the required levels for professional practice and compensate for current deficits in educational plans.

According to the recommendations elaborated by Margaret Mcallister and Jessica McKinnon [56], enhancing resilience in health professional education settings can be achieved through immersive learning experiences that focus on building work endurance, coping mechanisms, and developing strengths and leadership skills to navigate increased work demands. In addition, the learning and practice context should provide opportunities for reflection and assimilation of practice and peer knowledge. Increasing exposure to role models who teach adaptive strategies for success in healthcare settings is invaluable. Nurses who demonstrate resilience and post-traumatic growth can have a profound influence on aspiring students. It is important to engage in dialogue, critical analysis, and practical application of the valuable lessons learned from challenging situations. This approach should include collaborative decision-making training, reflective practice through clinical supervision, mentoring, or group support. Resilient nurses can motivate students by sharing their experiences through interactive methods such as seminars, conferences, or publications. Additional tools may include the establishment of cross-year forums for senior students to encourage critical thinking and constructive dialogue.

*4.2. Limitations*

One of the main strengths of this study is its longitudinal design over four years, which evaluates changes in resilience and affect during the bachelor's degree in nursing. The study had high participation and minimal loss of follow-up among subjects. However, we identified some limitations that should be considered. For instance, the sociodemographic information collected was limited to prevent the identification of students. Additionally, the sample selection was based on convenience, which may hinder the generalizability of the results. Finally, the categorization of sociodemographic variables, such as the over-representation of females in the sample, may lead to an underestimation or overestimation of the results. It is important to note that these limitations are inherent to our profession and the time of the study. In addition, this study was conducted in a single center, which may limit the generalizability of the main findings of this study. Future studies could replicate the design of this study on an international level to account for differences between different university programs and cultural influences. Also, adding a qualitative component to the design would enhance and extend the quantitative findings.

## 5. Conclusions

Nursing students experience an increase in resilience and positive affect, as well as a decrease in negative affect, throughout their 4-year training, despite events such as the COVID-19 pandemic in 2020. However, there is still room for improvement, particularly in the resilience levels of nursing students. These results must be considered as they would enable students to enter the job market with all the necessary adaptations, reducing the dropout rate and discomfort of nurses during their initial years of professional practice. Therefore, it is crucial to cultivate resilience during university studies. In addition, promoting peer and tutor support, teaching coping skills, and emphasizing the importance of spirituality could lead to more resilient professionals. This would compensate for current deficits in education plans.

**Author Contributions:** L.I.M.-S., conceptualization, methodology, writing—original draft preparation; A.M.-M., conceptualization, methodology, writing—original and editing; L.R.-L., data curation, visualization, software, validation; G.M., translation, validation, writing—reviewing and editing. All authors have made substantial contributions to all of the following: (1) the conception and design of the study, or acquisition of data, or analysis and interpretation of data, (2) drafting the article or revising it critically for important intellectual content, (3) final approval of the version to be submitted. All authors have read and agreed to the published version of the manuscript.

**Funding:** This research did not receive any specific grant from funding agencies in the public, commercial, or not-for-profit sectors.

**Institutional Review Board Statement:** The authors assert that all procedures contributing to this work comply with the ethical standards of the relevant national and institutional committees on human experimentation and with the Helsinki Declaration of 1975, as revised in 2008. The Ethics and Research Committee of the Faculty of Nursing, Physiotherapy and Podiatry approved all procedures involving human subjects/patients (approval number: FEFP 20/21, on 4 November 2020). All participants included in the study were informed, verbally and in writing, of the study objectives and conditions. Written informed consent was obtained from all participants.

**Informed Consent Statement:** Informed consent was obtained from all subjects involved in the study.

**Data Availability Statement:** The datasets generated and/or analyzed during the current study are not publicly available due to data protection policy but are available from the corresponding author upon reasonable request.

**Public Involvement Statement:** No public involvement in any aspect of this research.

**Guidelines and Standards Statement:** This manuscript was drafted against the STROBE for observational research.

**Acknowledgments:** We would like to thank the students for their generosity.

**Conflicts of Interest:** The authors declare that they have no competing interests.

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
