# Peer review of "An Exploration of Resilience and Positive Affect among Undergraduate Nursing Students: A Longitudinal Observational Study"

_nursrep, doi:10.3390/nursrep14020067_

Round 1

Reviewer 1 Report

Comments and Suggestions for Authors

Dear authors, congratulations on choosing the theme of your paper. Attached you will find a document with my suggestions and comments. Thank you for your time, wishing you a good worh week, Isabel Rabiais  

Reviewer 2 Report

Comments and Suggestions for Authors

Thank you for the opportunity to review this manuscript. It is an important issue facing nursing students on a global level. I appreciate the authors longitudinal design to explore these constructs. I offer the following suggestions to enhance the overall manuscript. 

1. Title: The use of the word "effectiveness" suggests a possible intervention which is not the case as described in the manuscript I am wondering if an alternate title might be considered "An Exploration of Resilience and Positive Affect Amoung Undergraduate Nursing Students: A Longitudinal Observation al Study. 

2. Introduction - in terms of the importance of resilience, could the authors add further details about the importance of resilience in the profession more broadly e.g. retention, ability to cope with job as a new graduate. Also wondering the authors could explain more clearly if the strategies outlined on p2 lines 59-65 have been incorporated into the specific university's curriculum. As written, it suggests this is across universities but not clear as to whether these were integrated in the specific curriculum.

Please clarify - was this study conducted in one university or across several universities. 

3. Under Methods: Please clarify if the instrument was completed in person or electronically. Please describe how participants were recruited (e.g., email, other media) and by whom? Were professors involved in the recruitment process. 

4. Study variables - could the authors provide the stem question in the description of the instruments. 

5. Results: Were any race-based/diversity characteristics collected as part of the demographic data? It is interesting that a large proportion of students lived with parents or relatives. This may differ in other countries/cities. 

6. Variations in Resilience - lines 137-139 - please clarify the statement beginning with "Of the students, 20.5%..." It is not clear what this refers to.

7. For Table 2 Some of the dimension scores are presented following the individual items. Suggest this be labelled more clearly. Why not report all of the dimensions? 

8. Table 3. This table could be condensed to reflect only the items reflecting statistically significant changes. 

9. Please see line 167 on page 6 - Looks like a sentence is missing a word. 

10. Discussion. The authors present some interesting findings reflected in the data. These are particularly important for faculty teaching in undergraduate programs internationally. I am wondering if the authors might add a paragraph explaining what they identify as the key education factors influencing resilience and affect. For example, which educational components/strategies  do students find most valuable? As this was not specifically as key variable - perhaps that could be included in a future study. 

11. Under limitations, to clarify - was this study conducted within one academic unviersity program? Adding a qualitative component to the design would enhance and expand on the quantitative findings. Again this could be considered for a future study. The study could be replicated at an international level to compare students in different countries. 

Reviewer 3 Report

Comments and Suggestions for Authors

As a nursing teacher, I found the article quite interesting, and it is clearly written and it progresses logically. The research has been carried out well and it answers the research questions that were set for it. Methodically properly implemented and the results properly presented. Ethics and reliability taken into account enough. The information base of the article is sufficient and it describes the situation in which nursing degrees are currently.

This research is particularly important because in several countries, probably around the world, there is information that nursing work no longer attracts young people like it used to, which nevertheless requires a wide range of tolerance, for example when dealing with other people, both patients and their relatives, as well as colleagues.

On the other hand, there are also indications that the new generation is not as resilient to adversity and the challenges of studying, work, and private life situations, like most likely was the were the older generations. That's why information is needed on how to develop the resilience question of health students.

University nursing education plays a crucial role in fostering resilience and positive affect among nursing students. Resilience, the ability to overcome adversity and learn from experiences, has a significant impact on learning, academic performance, course completion, and long-term professional practice. This study produced useful information about how the development of resilience develops during nursing education.

However, it did not produce much information about equipping nursing students with effective coping strategies, educational programs can help them resist stress and enhance their personal well-being. The requirements of nursing education are not sufficient to develop the level of sustainability required by future professional practice.

The results suggest that promoting support from colleagues and tutors, teaching coping skills and emphasizing the importance of spirituality can lead to more resilient professionals. This is important in order to reach the level required for professional practice and to compensate for the current gaps in training plans.

The shortcoming of this article is that it does not suggested by what means resilience could be developed in nursing education. I just missed this. There is mentioned that tutoring etc. could be added, but how, what way etc. was not mentioned.

Therefore, more research is needed on the relationship between nursing students' resilience and learning, as these qualities are vital to their academic success and future career continuity.
